# *Amburana cearensis* (Cumaru) and Its Active Principles as Source of Anti-Leishmania Drugs: Immunomodulatory Activity of Coumarin (1,2-Benzopyrone)

**DOI:** 10.3390/biomedicines13040979

**Published:** 2025-04-17

**Authors:** Naya Lúcia de Castro Rodrigues, Elizama Shirley Silveira, Francisco Rafael Marciano Fonseca, Ticiana Monteiro Abreu, Edilberto Rocha Silveira, Ana Bruna de Araújo, Maria Jania Teixeira, Luzia Kalyne Almeida Moreira Leal

**Affiliations:** 1Department of Pharmacy, Faculty of Pharmacy, Odontology and Nursing, Federal University of Ceará, Pastor Samuel Munguba Street, 1210, Fortaleza 60430-372, CE, Brazil; naya_castro@hotmail.com (N.L.d.C.R.); elizamassilveira@gmail.com (E.S.S.); rafaelmf.bio@gmail.com (F.R.M.F.); ana_bruna17@hotmail.com (A.B.d.A.); 2Department of Pathology and Legal Medicine, Federal University of Ceará, Monsenhor Furtado Street, w/n, Fortaleza 60441-750, CE, Brazil; ticiane.abreu@ufc.br; 3Department of Organic and Inorganic Chemistry, Science Center, Federal University of Ceará, Fortaleza 60430-900, CE, Brazil; edil@ufc.br

**Keywords:** *Amburana cearensis*, coumarin, *Leishmania braziliensis*, immunomodulatory action, cytokine, nitric oxide

## Abstract

**Background/Objectives**: In Brazil, *Leishmania braziliensis* is the main etiological agent of cutaneous leishmaniasis and represents an important public health problem. The actual pharmacotherapy of leishmaniasis has several disadvantages, making the development of new therapeutic options essential. The present study aimed to carry out the bioprospecting and selection of products of *Amburana cearensis,* including extracts and active principles with a leishmanicidal effect and to evaluate its possible mechanism of action. **Methods**: A dry extract of *A. cearensis* (DEAC) was characterized by HPLC, with the following active markers: coumarin (CM), amburoside A (AMR), and vanillic acid (VA). The leishmanicidal effect of DEAC was assessed, and the in vitro inhibitory action of the phenolic fraction, including CM, AMR, and VA, on promastigote and amastigote forms were determined. **Results**: CM showed the best reductions (maximal inhibition: 57%) of the promastigote form of *L. braziliensis*, followed by the plant extract (40% inhibition) and other test drugs (maximal reduction: 29%). The treatment of macrophages infected by *L. brasiliensis* with CM (10 μg/mL) reduced the intracellular parasite load (amastigote form, maximal reduction: 50%), increased the production of nitric oxide, TNF-α, IL-12, and IL-10, and decreased the production of IL-4. These effects were not related to cytotoxicity (MTT test). Glucantime (4 mg/mL, standard drug) reduced the amastigote form by 65%. **Conclusions**: CM showed promising leishmanicidal activity against both forms of *L. brasiliensis*, and this effect seems to be associated, at least in part, to its immunomodulatory action by tilting the Th1/Th2 imbalance in favor of Th1.

## 1. Introduction

Leishmaniasis is a parasitic infection caused by the protozoa of the genus *Leishmania* and transmitted by the bite of infected sand-flies [1,2]. The most common form of leishmaniasis is the localized cutaneous leishmaniasis (LCL), with approximately 0.7–1.3 million new cases each year [3]. In Brazil, *Leishmania braziliensis* is the main etiological agent of human cutaneous forms, becoming an important public health problem [4,5]. This species can cause clinical conditions with single or multiple ulcerated lesions and also leads to mucocutaneous involvement, with impairment of oropharyngeal mucosa, extensive tissue destruction, and high morbidity. In some cases, the parasite may even invade the bloodstream, resulting in disseminated cutaneous lesions [2,6]. Finally, the disease caused by *L. braziliensis* is particularly distinguished from other leishmaniasis by its chronicity, latency, and tendency to metastasis in the human host [7].

The clinical presentation of patients affected by leishmaniasis depends not only on the species of parasite but also on the immune response of the infected host [8,9]. The *Leishmania* replication control is associated with the development of a potent immune response mediated by T helper type 1 lymphocytes (Th1) producing tumor necrosis factor (TNF) α and interferon (IFN) γ. TNF-α promotes the parasite control by activating infected macrophages, which start to produce microbicidal agents, eliminating the parasites [7,9], whereas a response by T helper type 2 (Th2) favors the susceptibility to the disease, being characterized by the alternative activation of macrophages, through the production of several cytokines, including interleukins (IL) 13, 5, and, mainly, 4 [10].

Some drugs are being currently used for the treatment of leishmaniasis, including the pentavalent antimonials, amphotericin B and miltefosine; however, they have several disadvantages, such as high toxicity, long administration schedules, and the resistance that some parasites already have to these drugs [1,11]. Therefore, the development of new medicines with greater efficiency and less side effects is necessary. In this context, medicinal plants emerge as potential source, such as *Amburana cearensis* (Allemao) A.C. Smith, which belongs to the Fabaceae family and has been widely used in traditional medicine in the Caatinga of Northeast Brazil in Brazilian popular medicine [12,13,14,15].

Previous studies of *A. cearensis* allowed for the description of its chemical characteristics, and among the main constituents of this plant are coumarin (CM) (1,2-benzopyrone), phenol glucosides (e.g., amburoside A-AMR), flavonoids, and phenol acids [13]. A few years ago, our laboratory showed the antineuroinflammatory effect of *A. cearensis* and its active principles (CM and AMR) expressed by the reduction in inflammatory mediators, which seems to occur by inhibition of the JNK and ERK1/2 MAPKs pathways [16]. Our group and other laboratories also determined other pharmacological properties of *A. cearensis* and/or its chemical constituents, including antioxidant [17,18], muscle relaxant [19], antimicrobial [20,21], and antimalarial activities [22]. Bravo et al. [23] determined the in vitro effects of CM against *L. amazonensis*, *L. braziliensis*, and *L. donovani*. Although there are already studies using this compound to control *L. braziliensis*, these works were carried out using only the promastigote form, without investigating its mechanism of action.

Therefore, considering the impact of leishmaniasis, especially in developing countries, the limitations imposed by actual pharmacotherapy, and the therapeutic potential of *A. cearensis* and its chemical constituents, the present study aimed to carry out the bioprospecting and selection of active materials from *A. cearensis* with a leishmanicidal effect (in vitro and in vivo) against *L. braziliensis*, including the evaluation of possible immunomodulatory action.

## 2. Materials and Methods

### 2.1. Botanical Material

Trunk bark of *Amburana cearensis* (Allemão) A.C. Sm. was collected at the São Vicente farm, in the city of Quixeramobim, Ceará, Brazil. Voucher specimens (nº. 837 and 847) were deposited at the Prisco Bezerra Herbarium, located in the Department of Biology of the Federal University of Ceará.

### 2.2. Chemicals

Amburoside A (AMR) was isolated from *A. cearensis* according to the methodology described previously by our group [24]. Vanillic acid (VA), coumarin (CM), 3-(4,5-Dimethylthiazol-2-yl)-2,5-Diphenyltetrazolium Bromide (MTT), and dimethyl sulfoxide (DMSO) were obtained commercially from Sigma-Aldrich, St. Louis, MO, USA.

### 2.3. Preparation and Chemical Characterization of Dried Extract from Amburana Cearensis (DEAC)

The extract of *A. cearensis* was prepared by maceration of the trunk bark with ethanol, in a 1:1 ratio, according to the methodology described by Araruna *et al.* [25]. Subsequently, the dried extract of *A. cearensis* (DEAC) was prepared using a mini spray-dryer (Labmaq do Brazil Ltd., Ribeirão Preto, Brazil). The colloidal silicon dioxide was used as a drying carrier (30% of the solid residues) with a feed flow of 1 L/h, inlet temperature of 100 °C, and airflow of 40 L/min. The chemical characterization was performed using a high-performance liquid chromatography–photodiode array detector (HPLC-PDA) (Waters, Milford, MA, USA) according to the validated method developed previously by our laboratory [24]. HPLC-PDA analysis of DEAC allowed for the determination of the chromatography profile of plant extract and the quantification (mg/g of dried extract) of three bioactive markers, AMR (48.57 ± 0.267 mg/g), CM (15.12 ± 1.118 mg/g), and VA (1.61 ± 0.006 mg/g).

### 2.4. Amburana Cearensis Phenolic Fraction (ACPF) Obtainment and Characterization

The *A. cearensis* methanolic extract (ACME) was obtained after extraction in a Soxhlet system, filtration, and low-pressure rotary evaporation. After submitting an aliquot of the ACME to a solid–liquid partition with distilled water, a water-soluble fraction and a residual fraction (ACME/RES) were obtained. ACME/RES was initially submitted to extraction in a Soxhlet system with hexane, then with dichloromethane, and, later, with ethyl acetate. Finally, the residue present in the cartridge was extracted exhaustively in methanol. The solutions were dried in a rotary evaporator and resulted in the fractions ACME/RES-H (2.76 g), ACME/RES-D (5.84 g), ACME/RES-Ac (1.01 g), and ACME/RES- M (21.43 g), respectively. The 1H NMR spectrum of the AC-ME/RES-M fraction showed characteristic signs of coumarin being subjected to a new exhaustive extraction with dichloromethane in a Soxhlet system, giving rise to the dichloromethane fraction, ACME/RES-MD (3.88 g), and the residual fraction, ACME/RES-MR (15.43 g). The latter presented, mostly, in its 1H NMR spectrum, characteristic signs of amburoside A, which was referred to as ACPF and used for pharmacological tests.

### 2.5. Parasites

The *Leishmania braziliensis* strain (THOR/MCAN/BR/98/R619) was provided by Dr. Alda Maria da Cruz from Fundação Oswaldo Cruz (Fiocruz/Rio de Janeiro, RJ, Brazil) and was isolated from a patient with cutaneous leishmaniasis. Parasites were cultured at 25 °C in Neal, Novy and Nicolle medium (NNN; 10% blood agar), and Schneider’s insect medium, pH 7.0, supplemented with 10% fetal bovine serum (FBS), 2% sterile human urine, and 2% antibiotics (10,000 U/mL penicillin and 10 mg/mL streptomycin), all from Sigma-Aldrich, São Paulo, SP, Brazil. For the experiments, the parasites were used until the fifth in vitro passage. The promastigotes were subjected to three wash cycles with ice-cold sterile saline (NaCl 0.9%), centrifuged (2000× *g*; 15 min; 4 °C), and adjusted with culture media to the desired concentrations in each experiment.

### 2.6. Leishmanicidal Effect Evaluation

Initially, a screening was performed with the following *A. cearensis* compounds: ethanolic extract (DEAC), phenolic fraction (ACPF), AMR, CM, and VA. CM and VA were commercially obtained (Sigma-Aldrich). For the experiments, these compounds were diluted in 0.1% dimethyl sulfoxide (DMSO; VETEC, Rio de Janeiro, RJ, Brazil), and those with the best action were used in the subsequent tests.

#### 2.6.1. Effect Evaluation Against the Promastigote Forms

A culture containing the *L. braziliensis* promastigote forms was centrifuged (2000× *g*; 15 min; 4 °C), resuspended in supplemented Schneider medium, counted in a Neubauer chamber, and diluted in the same medium, to obtain a concentration of 10^7^ promastigotes/mL. Subsequently, promastigotes were distributed in 48-well plates (160 μL/well), and 40 μL of DEAC, ACPF, AMR, CM, and VA were added, for final concentrations of 10, 25, 50, and 100 μg/mL. As a control treatment, DMSO (0.1%), amphotericin B (AMB) (16 µg/mL; Sigma-Aldrich), and supplemented Schneider medium (Sigma-Aldrich, St. Louis, MO, USA) were used. Plates were incubated (25 °C) and, after 24 and 48 h, the promastigotes’ viability was evaluated by counting in a Neubauer chamber using Trypan blue solution (0.2%). All experiments were performed in triplicate, and the results are expressed as the survival rate (%).

#### 2.6.2. Cytotoxicity Assessment

Macrophages of the RAW 264.7 lineage were obtained from the cell bank of the Federal University of Rio de Janeiro (BCRJ-UFRJ) and grown in microplates containing RPMI 1640 medium (Sigma-Aldrich) supplemented with 10% FBS and 2% antibiotics (10,000 U/mL penicillin and 10 mg/mL streptomycin). For each experiment, the macrophage culture was centrifuged (500× *g*; 15 min; 5 °C), resuspended in supplemented RPMI medium, counted in a Neubauer chamber, and diluted in the same medium, to obtain a concentration of 5 × 10^5^ cells/mL.

For the experiments, 200 µL/well of this suspension were placed in sterile 96-well plates and incubated for 16 h (37 °C and 5% CO_2_). Subsequently, 20 µL of medium were removed from the wells, except for viability controls, and 20 µL of DEAC and CM were added, for final concentrations of 10, 25, 50, and 100 µg/mL. Then, the plate was incubated (37 °C; 5% CO_2_) for 24 and 48 h. After this period, plates were washed with 200 µL/well of sterile phosphate buffer saline (PBS; Sigma-Aldrich), pH 7.4, at 37 °C; 200 µL/well of supplemented RPMI was added, and the plate was incubated for 90 min (37 °C; 5% CO_2_). Then, 100 µL/well were removed, and 100 µL/well of MTT (3-(4,5-dimethylthiazol-2-yl)-2,5-diphenyltetrazolium bromide), at a concentration of 1 mg/mL in sterile PBS, was added, and the plate was incubated for 4 h (37 °C; 5% CO_2_). Finally, MTT was removed, and 150 µL/well of pure DMSO were added, and the plate was wrapped in aluminum foil, shaken for 10 min, and allowed to rest for 3 min. The absorbance reading was performed in a microplate at 570 nm. As control treatment, supplemented RPMI, pure DMSO, and 0.1% and glucantime^®^ (4 mg/mL) were used.

Cell viability of the samples was calculated by the following equation: % inhibition = (A_control_ − A_substance_/A_control_ × 100), in which A represents the absorbance of the samples. The experiments were performed in triplicate, and the results are expressed as the cell viability rate (%).

#### 2.6.3. Effect Evaluation Against the Amastigote Forms

Macrophages of the RAW 264.7 lineage were centrifuged (500× *g*; 15 min; 5 °C), resuspended in supplemented RPMI medium, counted in a Neubauer chamber, and diluted in the same medium, to a concentration of 1 × 10^6^ cells/mL. Then, in 24-well plates containing round glass coverslips (23 mm) at the bottom, macrophages were distributed (10^6^ cells/coverslip) and incubated for 24 h (37 °C; 5% CO_2_). Non-adhered cells were removed by washing with RPMI medium and cultured with this medium supplemented in the absence or presence of *L. braziliensis* (10^7^ promastigotes/mL), in a ratio of 10 parasites to 1 macrophage, for 12 h. After this period, wells were washed with RPMI medium at 37 °C, received CM solutions at final concentrations of 10, 25, 50, and 100 µg/mL, and plates were incubated at 37 °C and 5% CO_2_. As a control group, supplemented RPMI medium, DMSO (0.1% in water, drug vehicle), and glucantime^®^ (GLU; 4 mg/mL) were used. After 24 and 48 h, the supernatants were collected and stored in a freezer at −80 °C for further nitric oxide and cytokines dosages.

For the macrophages’ parasite load analysis, after the incubations, the coverslips were removed from each well, washed with sterile saline (NaCl 0.9%), fixed, stained with Giemsa (Sigma-Aldrich), and glued, after drying, on glass slides. Evaluation was carried out under an optical microscope (100×). A total of 50 macrophages were examined, and the number of amastigotes/50 macrophages was determined. Experiments were performed in triplicate, and the results are expressed in amastigotes/50 macrophages.

##### Measurement of Nitric Oxide (NO) and Cytokines

NO production was measured by the dosage of its degradation product, nitrite, using the colorimetric method with Griess reagent [26]. To this end, 100 μL of the supernatant was incubated with 100 μL of the Griess reagent, which consisted of equal parts (1:1:1:1) of phosphoric acid 5%, sulfonylamide dissolved in phosphoric acid 5%, N-(1-naphthyl)-thylenediamine dihydrochloride 0.1%, and distilled water at room temperature for 10 min. Absorbance was measured at 560 nm in a microplate reader. Nitrite content was determined from a sodium nitrite standard curve, and the results were analyzed using the Softmax PRO program (Molecular Devices, Sunnyvale, CA, USA) and expressed as μmol nitrite/mL.

The levels of cytokines IL-4, -10, and -12 and TNF-α were determined by the enzyme-linked immunosorbent method (ELISA), using commercial kits, following the manufacturer’s recommendations (BD Biosciences, San Jose, CA, USA). Results were also analyzed using the Softmax PRO program and expressed as pg/mL.

### 2.7. Statistical Analysis

Results are presented as the mean ± S.E.M. (standard errors of the mean) and were compared by regular one-way ANOVA followed by the Tukey test. The statistical program used was GraphPad Prism 7.0 Version for Windows, GraphPad Software 7.0 Version (San Diego, CA, USA). *p* < 0.05 was considered statistically significant.

## 3. Results

### 3.1. Anti-Leishmanial Activity of Dried Extract, Phenolic Fraction, and Molecules of A. cearensis in L. braziliensis Promastigotes

DEAC did not reduce the promastigotes’ survival after 24 h of incubation; however, after 48 h, at concentrations of 25, 50, and 100 μg/mL, it significantly decreased their survival by 30.87% ± 4.66, 38.49% ± 5.98, and 40.44% ± 3.97 (Figure 1A,B). ACPF significantly lowered the promastigotes survival only at the concentration of 100 μg/mL by about 29% (Figure 1C,D). AMR (25, 50, and 100 μg/mL) diminished promastigotes survival only after 48 h of incubation by 23.8% ± 2.1, 29.4% ± 1.7, and 23.7% ± 3.3, respectively (Figure 1E,F). VA (25 and 50 μg/mL) also significantly reduced the survival only after 48 h, by 26.0% ± 2.5 and 28.6% ± 2.8, respectively (Figure 1G,H). Finally, CM (25, 50, and 100 μg/mL), after 24 h of incubation, significantly decreased the promastigote form’s survival, by 31.4% ± 8.5, 22.7% ± 8.2, and 21.5% ± 8.8, respectively; and, after 48 h, at all concentrations, CM significantly reduced this survival by 27.6% ± 6.3, 31.8% ± 4.3, 39.9 ± 5.8, and 57.2% ± 3.5, respectively (Figure 1I,J). DEAC and CM presented the most promising results, which is why they were chosen for further studies.

### 3.2. Cytotoxicity Assessment of DEAC and CM in Macrophages

The addition of increasing concentrations of DEAC (10, 50, and 100 μg/mL) in macrophages reduced significantly the viability of cells by 28.7% ± 6.6, 41.7% ± 2.9, and 65.7% ± 2.0, respectively, only after 24 h incubation (Figure 2A). However, DEAC was not cytotoxic for cells when incubated for 48 h (Figure 2B). CM (10, 50, and 100 μg/mL) did not significantly decrease cell viability at any of the observed times (Figure 2C,D). Glucantime—GLU (4 mg/mL)—the main drug for the cutaneous leishmaniasis treatment in Brazil, was cytotoxic to macrophages, significantly diminishing the viability of these cells by 68.5% ± 9.16 and 93.1% ± 0.69 after 24 and 48 h of incubation, respectively, when compared to 0.1% control treatment (DMSO). Thus, based on the potential anti-leishmanial activity of CM, which was not toxic to macrophages, this compound was chosen to continue the studies.

### 3.3. Effect of CM on Leishmania braziliensis Amastigotes

Following the study described above, we investigated the effect of CM on the parasite load in macrophages infected by *L. braziliensis*, including a morphological evaluation. The treatment of macrophages with CM (time of incubation: 24 h), at concentrations of 10, 25, and 50 μg/mL, caused a significant reduction in parasite load (50.95 ± 8.17, 57.20 ± 12.13, and 51.40 ± 2.52 amastigotes/50 macrophages, respectively), when compared to the control group treated with 0.1% DMSO (vehicle; 100 ± 4.67 amastigotes/50 macrophages) (Figure 3). These results obtained by treatment with CM (10 to 50 μg/mL) did not differ significantly when related to GLU (35.02 ± 2.77 amastigotes/50 macrophages). CM, when incubated for a longer period (48 h), was able to decrease the parasite load only at higher concentrations (50 and 100 μg/mL) (Figure 3). The treatment of macrophages with CM did not alter their morphological characteristics, preserving the integrity of the plasma membrane and nucleus when related to untreated macrophages. The macrophages infected by *L. braziliensis* exhibited vacuoles in their cytoplasm, while macrophages not infected did not show this morphological alteration (Figure 3A). Both GLU (Figure 3C) and CM (Figure 3D) reduced the number of infected cells with vacuoles in their cytoplasm.

#### CM Induces an Increase in Nitric Oxide Production and Modulates the Cytokine Profile in Macrophages Infected by *L. braziliensis*

To evaluate the role of nitric oxide and cytokines in mediating the leishmanicidal effect of CM in macrophages infected with *L. braziliensis,* the concentration of these biomarkers (nitrite, IL-4, IL-10, IL-12, and TNF-α) in the absence and presence of CM in infected cells was measured. The infection of cells by *L. braziliensis* reduced significantly the nitrite production (0.1% DMSO group: 5.04 ± 0.90 pg/mL). However, the treatment of infected cells with CM (10, 25, 50, and 100 µg/mL) after 48 h of incubation resulted in a significant rise in nitrite levels at all concentrations tested (42.71 ± 0.01, 40.62 ± 2.50, 70.94 ± 6.81, and 114.20 ± 0.01 pg/mL, respectively) when related to the 0.1% DMSO group. The treatment of infected cells with GLU (4 mg/mL, reference drug) resulted in an augmentation of more than 40-fold in the nitrite level (229.40 ± 15.70 pg/mL) when related to the 0.1% DMSO group (5.044 ± 0.9011 pg/mL). Coumarin after 24 h of incubation in infected macrophages did not induce a significant increase in nitrite levels when compared to the 0.1% DMSO group, while GLU (7.618 ± 1.058 pg/mL) resulted in an increase of only about 39,7% in the nitrite level when related to the 0.1% DMSO group (5.454 ± 0.2147 pg/mL) (Figure 4).

According to the cytokines evaluated, the treatment of infected cells with CM lead to an increase in the production of cytokines (after 24 and/or 48 h of incubation), one of the main targets in leishmanicidal action. Upon the infection of macrophages by *L. braziliensis*, both CM and GLU increased the production of IL-12 in infected macrophages when related to the 0.1% DMSO group (vehicle/control group). The highest concentrations of CM after 24 h of incubation (50: 40.16 ± 8.801 pg/mL; 100: 37.81 ± 6.00 pg/mL) showed better results than GLU (4 mg/mL: 27.36 ± 1.266 pg/mL) when compared to the 0.1% DMSO group (5.474 ± 0.757 pg/mL). After 48 h of incubation of CM or GLU in infected macrophages, only GLU and CM (50 µg/mL) augmented the concentration of IL-12 (9.04 ± 1.24 and 10.51 + 0.11 pg/mL, respectively), in relation to the DMSO group (3.33 ± 1.55 pg/mL). There was no statistical difference between the GLU and CM groups (Figure 5A,B).

The infection of macrophages by *L. braziliensis* induced a significant reduction in the production of TNF-α, which was reversed by CM in a concentration-dependent manner after 24 h of incubation with this test drug. The maximal effect of CM was observed when it was incubated for 48 h, achieving maximal effect at higher concentration (pg/mL) (50: 146 ± 4.01 pg/mL; 100: 124 ± 14.24 pg/mL) when related to the 0.1% DMSO group (3.575 ± 0.182 pg/mL). It is important to note that GLU, with a concentration 40× higher than that of CM (µg/mL), did not significantly alter the levels of TNF-α in any of the analyzed times (24h: 13.65 ± 0.46 pg/mL; 48h: 14.70 ± 0.48 pg/mL) when compared to the 0.1% DMSO group (Figure 5C,D).

Infection induced a significant increase in IL-4 levels only after 24 h of evaluation (24 h: 45.56 ± 4.987 pg/mL) when compared to RPMI group (24 h: 16.36 ± 2.089 pg/mL; 48 h: 29.18 ± 4.988 pg/mL). This effect was reversed by CM after 24 h of incubation with infected cells, reducing by up to 78% the level of IL-4 compared to the 0.1% DMSO group, while GLU (4 mg/mL) augmented the IL-4 level in infected cells by only around 18% (Figure 5E,F).

Finally, after 24 h of incubation with infected macrophages, CM, at concentrations of 25, 50, and 100 µg/mL, and GLU (4 mg/mL) significantly increased IL-10 levels (15.49 ± 1.50, 19.70 ± 0.91, 35.96 ± 4.04, and 17.30 ± 1.08 pg/mL, respectively), when compared to the DMSO group (5.47 ± 0.76 pg/mL) (Figure 5G). However, after 48 h of incubation of infected macrophages with CM or GLU, only CM (50 and 100 µg/mL) maintained the high levels of this cytokine (37.18 ± 8.83 and 44.34 ± 11.55 pg/mL, respectively), in relation to the DMSO group (1.26 ± 0.21 pg/mL) (Figure 5H). It is noteworthy that these two CM concentrations also differed significantly from the reference drug GLU (0.42 ± 0.08 pg/mL).

## 4. Discussion

In the present study, the leishmanicidal effect of the standardized dried extract, phenolic fraction, and active principles (CM, AMR, and VA) of *A. cearensis* were evaluated. Coumarin (CM) exhibited higher growth inhibition of the promastigote form of *L. braziliensis*, followed by DEAC and other test drugs. The findings also showed that CM had antileishmanial activity against the amastigotes form of *L. braziliensis*, which seems to be associated with regulation of the production of key markers for the control of leishmaniasis.

It is important to emphasize that the result obtained for CM (10–100 µg/mL) regarding activity against promastigotes corroborates the study carried out by [27], in which the in vitro antileishmanial activity of this metabolite was reported against promastigote forms of *Leishmania* sp. Other types of coumarin were also effective in vitro against promastigote forms of *Leishmania*. The compound 7-geranyloxycoumarin, belonging to the class of coumarins and known as auraptene, isolated from the leaves of *Esenbeckia febrifuga*, as well as the coumarins from *Ferula szowitsiana* roots, showed inhibition of the growth of promastigotes of *L. major* [28,29]; while coumarins isolated from *Platymiscium floribundum* showed leishmanicidal activity after 24 h against *L. donovani*, *L. mexicana*, and *L. major*, at a concentration of 100 μg/mL [30].

It is worth mentioning that the research carried out on promastigote forms does not represent an infection of the mammalian host, since, in the natural life cycle of *Leishmania*, these flagellated forms are present in the insect vector [31,32]. Promastigotes differ significantly from amastigotes in terms of morphology, surface glycocalyx composition, and metabolism [33,34,35]. In a study using Ascaridole, [1-methyl-4-(1-methylethyl)−2,3-dioxabicyclo[2.2.2]oct-5-ene], a bicyclic monoterpene endoperoxide present in *Chenopodium ambrosioides L.* (Amaranthaceae), against *L. donovani* promastigote forms, this compound inhibited the glycolysis of parasites with ATP depletion, which culminated in cell death similar to apoptosis [36]. The effect of DEAC and its active principle on the promastigote form of *L. braziliensis* is, therefore, important for aiding in the control of disease transmission, and studies are planned to clarify the mechanism of action.

Although the promastigote form is easier to handle, it has the limitation of presenting some metabolic differences with the amastigote form, which is intracellular and infective to mammals, and these differences can alter the response to drugs [37]. Furthermore, compounds that show activity against promastigotes may be unable to reach the amastigote forms of *Leishmania* sp., as they could be incapable of crossing the macrophage membrane or remain stable with the low pH within these cells [38]. Therefore, experimental models based on activity against amastigote forms are also essential, since this is the parasite life stage responsible for the different clinical manifestations [39].

DEAC and CM showed better effects in reducing the survival of the promastigote form of *L. braziliensis* compared to other active compounds from the plant. However, only CM was not cytotoxic to macrophages. Thus, we decided to continue the studies on the amastigote form of *L. braziliensis* with CM, which is the main active principle of *A. cearensis*.

In the present work, CM demonstrated an anti-amastigote effect at both analyzed times (24 and 48 h). Corroborating our data, (-) mammea A/BB, a coumarin from *Calophyllum braziliensis*, at a concentration of 80 μg/mL, reduced the amastigote forms of *L. braziliensis* by 67.5% after 24 h of incubation [40]. Furthermore, osthole (osthol), a coumarin from Prangos asperula, decreased the amastigote viability of *L. major* after 72 h, at a concentration of 50 µg/mL [41]. The mechanism of leishmanicidal action of coumarins has not yet been fully elucidated, but their ability to increase the phagocytic activity of macrophages is well known, which may be the possible mechanism of action of their anti-amastigote action [42].

It is noteworthy that, although the treatment with GLU reduced the intracellular parasites number in relation to the control group, the concentration of this drug was 40× higher than the highest CM one (100 μg/mL) used in this study, demonstrating the superior effect of this compound. In addition, glucantime (GLU) was cytotoxic at all times analyzed, unlike CM. These data are very important considering that GLU is used by the Brazilian Unified Health System for the treatment of patients with leishmaniasis.

A good candidate for an antileishmanial agent should be one that stimulates macrophages to produce antimicrobial molecules such as nitric oxide (NO) and reactive oxygen species (ROS), which are required to effectively eliminate intracellular amastigotes [43,44]. Macrophages are the main host cells of *Leishmania* sp. and have a relevant role in the immunological control of these intracellular parasites, since, when activated by cytokines, they can generate large amounts of NO, which is the main effector molecule in the death of amastigotes [7,45,46]. Parasites of the genus *Leishmania* compromise the metabolism of macrophages, impairing their activation, thus preventing the production of NO, which facilitates their survival [7]. In this way, the effect of CM on the production of NO by macrophages infected with *L. braziliensis* was evaluated. The CM was able to induce NO production after 48 h on infected macrophages, which is consistent with the induction kinetics of induced nitric oxide synthase (iNOS), which acts between 48 and 72 h after infection [47].

The outcome of *L. braziliensis* infection depends on the type of immune response that the host develops. The type 1 helper T cell (Th1) response favors resistance to the parasite, whereas the type 2 helper T cell (Th2) response promotes susceptibility [38]. In parasite resistance, the production of IL-12 by macrophages and dendritic cells leads to the production of IFN-γ by NK cells and differentiation of Th0 cells into Th1, which will also produce IFN-γ, consequently providing a key stimulus for the development of macrophage resistance, with the production of reactive oxygen and nitrogen species by these cells [47,48]. On the other hand, susceptibility to *Leishmania* is generally supported by the early production of IL-4, which promotes the development of lesions during the early stages of infection [49], and this cytokine is able to downregulate IL-12 production [8,50,51]. Furthermore, the infection of macrophages by *Leishmania* sp. can lead to the production of IL-10, which deactivates macrophages [52], thus inhibiting the respiratory burst and the inflammatory cytokines’ production, especially TNF-α. It is worth mentioning that the role of IL-10 in human cutaneous leishmaniasis caused by *L. braziliensis* is not so clear, and it has already been suggested that the absence of this cytokine could be the cause of the worsening of the disease, due to the exaggerated inflammation caused by Th1 cytokines [53]. Therefore, in these cases, this cytokine has a regulatory role [54].

In the present study, it was observed that CM performs its anti-amastigote activity through a modulation of the inflammatory response, since this metabolite induced an increase in the production of TNF-α, IL-12, and IL-10 and a decrease in IL-4 secretion. It is noteworthy that, 48 h post-infection, there was a sharp decrease in IL-4 levels, an increase in IL-10, and a reduction in IL-12 induced by CM, highlighting, therefore, the ability of this molecule to balance the Th1 and Th2 responses. The literature has shown that the essential prerequisites for an effective immunomodulatory and leishmanicidal compound should be its potential to activate phagocytic cells, mainly macrophages, and tilt the Th1/Th2 imbalance in favor of Th1 [43,55], mechanisms that were observed by CM in the present study. CM emerges as a promising molecule for leishmanicidal activity (*L. braziliensis*) in both forms of the parasite, with lower toxicity compared to glucantime (reference leishmanicidal drug). Therefore, it is important to continue studies to understand how coumarin modulates the production of markers such as cytokines and the possible mechanism of cell death in the parasite.

## 5. Conclusions

The present study found that 1,2 benzopyrone (CM), an active pharmaceutical ingredient of *Amburana cearensis*, has a leishmanicidal effect against *Leishmania braziliensis* (promastigote and amastigote forms), without showing toxicity to macrophages. The leishmanicidal effect (promastigote form) of a dry extract from *A. cearensis* (DEAC) might be due to CM, while the relative toxicity of DEAC to macrophages could be caused by another chemical constituent of *A. cearensis*. Nevertheless, it still needs to be proven whether the leishmanicidal effect of DEAC is indeed due to CM or not. In an unprecedented manner, it was demonstrated that CM exerts its leishmanicidal effect through its ability to activate macrophages modulating the inflammation in favor of Th1 response, which promotes parasite death. These findings, therefore, characterize the active principle of *A. cearensis* CM, as a promising pharmaceutical ingredient for the treatment of leishmaniasis.

## Figures and Tables

**Figure 1 biomedicines-13-00979-f001:**
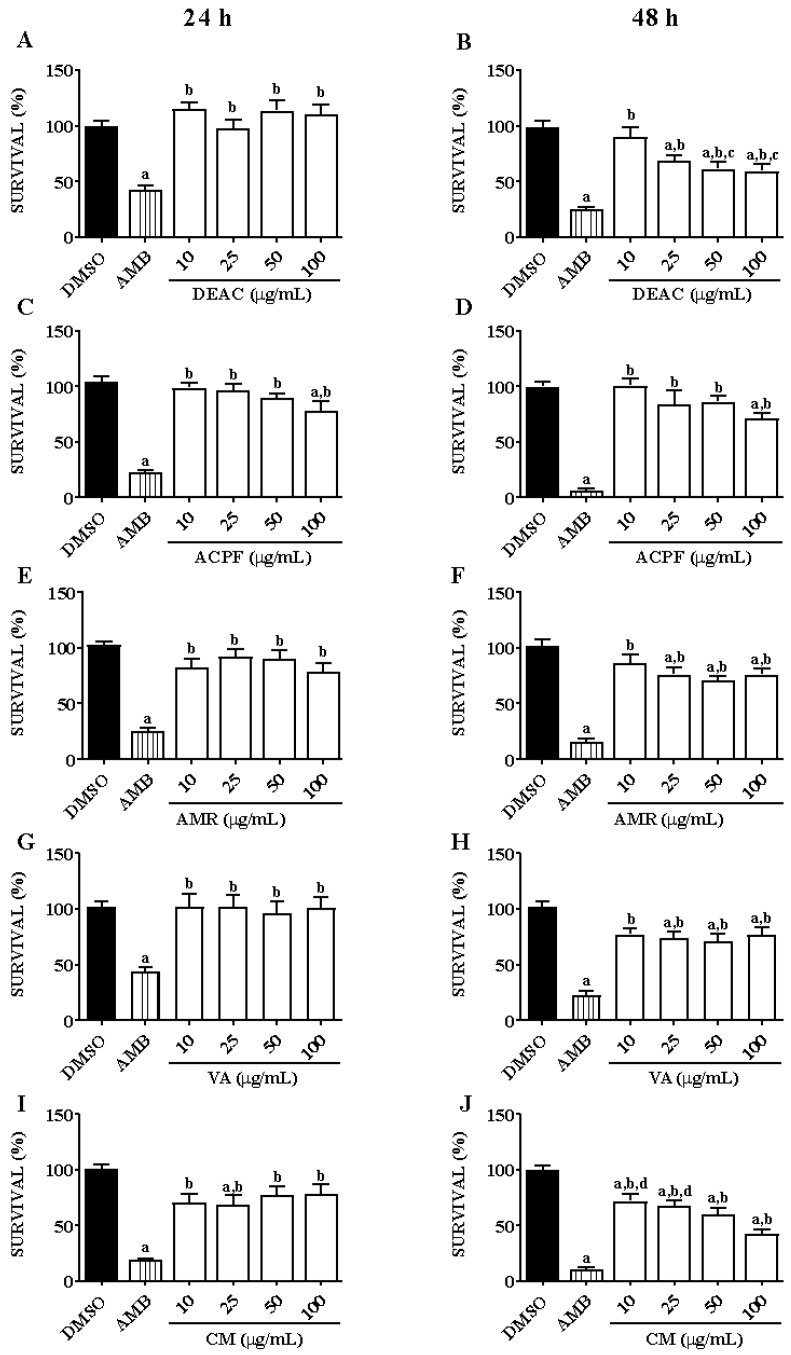
Effect evaluation against the promastigote forms. Promastigotes (10^7^ parasites/mL) were incubated with DMSO (0.1%; negative control), AMB (16 µg/mL; positive control), DEAC, ACPF, AMR, VA, and CM (10, 25, 50, and 100 µg/mL) for 24 and 48 h. All experiments were performed in triplicate, and the results are expressed as the survival rate (%). AMB: amphotericin B; DEAC: dried extract of *A. cearensis*; ACPF: *A. cearensis* phenolic fraction; AMR: amburoside A; VA: vanillic acid; and CM: coumarin. Each column represents the mean ± S.E.M. *p* < 0.05 to DMSO (a); to AMB (b); to 10 (c); and to 100 (d) (one-way ANOVA; Tukey test). (**A**) Incubation with DEAC for 24 h, (**B**) Incubation with DEAC for 48 h, (**C**) Incubation with ACPF for 24 h, (**D**) Incubation with ACPF for 48 h, (**E**) Incubation with AMR for 24 h, (**F**) Incubation with AMR for 48 h, (**G**) Incubation with VA for 24 h, (**H**) Incubation with AMR for 48 h, (**I**) Incubation with CM for 24 h, (**J**) Incubation with CM for 48 h.

**Figure 2 biomedicines-13-00979-f002:**
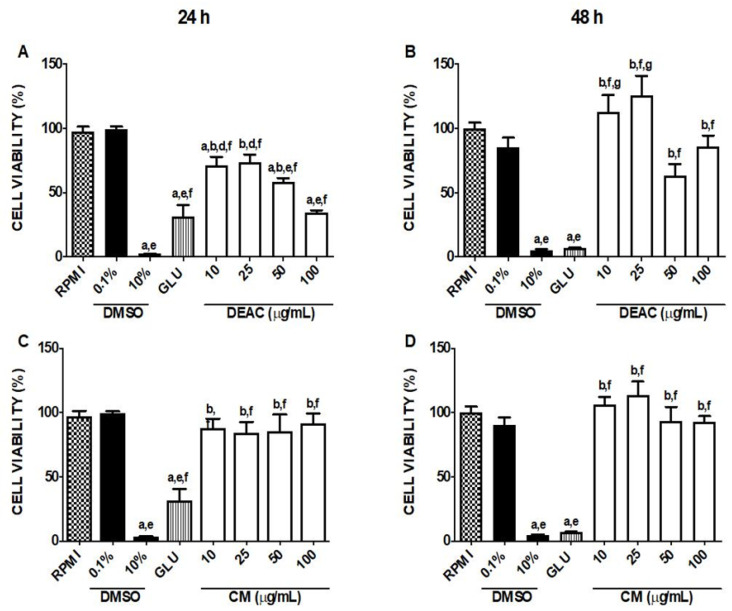
Cytotoxicity assessment. Macrophages of the RAW 264.7 lineage (5 × 10^5^ cells/mL) were incubated (37° C; 5% CO_2_) with DEAC and CM (10, 25, 50, and 100 µg/mL), and, after 24 and 48 h, the cell viability was evaluated using the MTT test. As a control group, supplemented RPMI, DMSO 10% (cytotoxic standard), and 0.1% and GLU (4 mg/mL) were used. All experiments were performed in triplicate, and the results are expressed as the cell viability rate (%). GLU: glucantime^®^; DEAC: ethanolic extract; and CM: coumarin. Each column represents the mean ± S.E.M. *p* < 0.05 to RPMI (a); to 0.1% DMSO (b); to GLU (d); to 10 (e); to 10 (f); to 50 (g) (one-way ANOVA; Tukey test). (**A**) Cell viability with DEAC 24 h, (**B**) Cell viability with DEAC 48 h, (**C**) Cell viability with CM 24 h, (**D**) Cell viability with CM 48 h.

**Figure 3 biomedicines-13-00979-f003:**
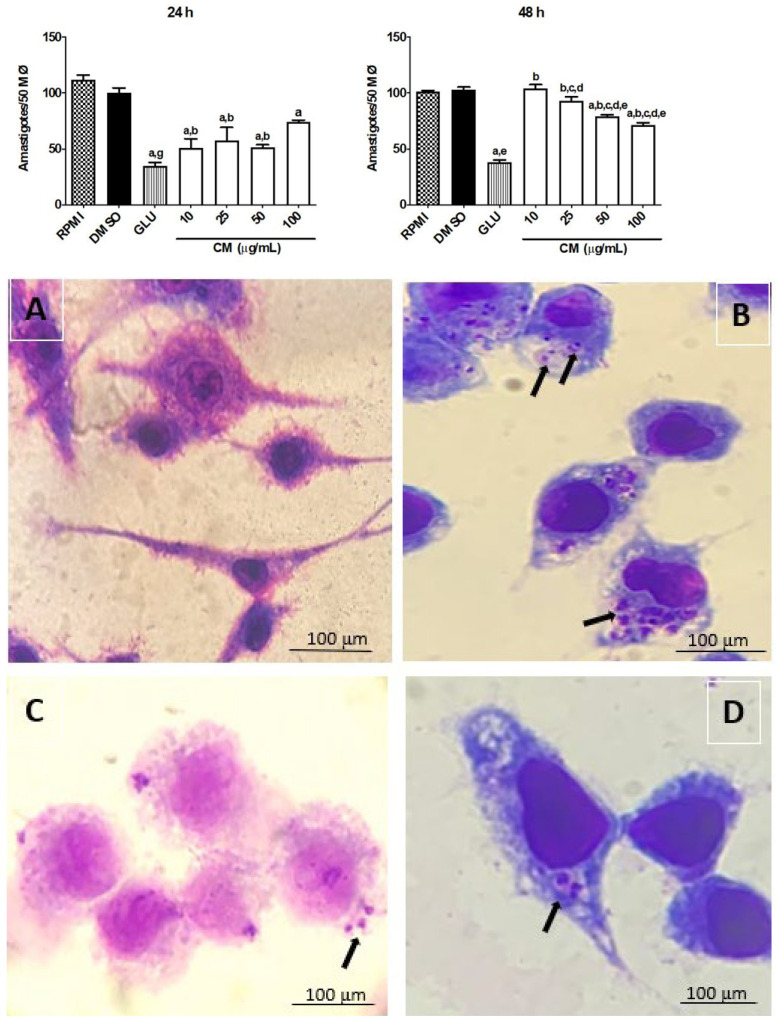
Effect evaluation against the amastigote forms: macrophages of the RAW 264.7 lineage (1 × 10^6^ cells/mL) were cultured with *L. braziliensis* (10^7^ promastigotes/mL), in a ratio of 10 parasites to 1 macrophage. After 12 h, they were incubated (37 °C; 5% CO_2_) with CM (10, 25, 50, and 100 µg/mL) for 24 and 48 h. As a control, supplemented RPMI medium, DMSO 0.1%, and GLU (4 mg/mL) were used. Evaluation was carried out under an optical microscope (100×), and fifty macrophages were examined. Experiments were performed in triplicate, and the results are expressed in amastigotes/50 macrophages. GLU: glucantime^®^, and CM: coumarin. Each column represents the mean ± S.E.M. *p* < 0.05 to RPMI (a); to 0.1% DMSO (b); to GLU (c); to CM 10 µg/mL; (d); and CM to 25 µg/mL (e) and to 100 (g) (one-way ANOVA; Tukey test). Photomicrograph of RAW 264.7 lineage macrophages infected with *L. braziliensis* and treated with CM. (**A**): uninfected macrophages; (**B**): macrophages infected + 0.1% DMSO (vehicle); (**C**): macrophages infected and treated with GLU (4 mg/mL; positive control); and (**D**): macrophages infected and treated with CM (100 µg/mL). Cells were stained with Giemsa, and the evaluation was carried out under an optical microscope (100×). The arrows indicate the presence of *L. braziliensis* in the amastigote form.

**Figure 4 biomedicines-13-00979-f004:**
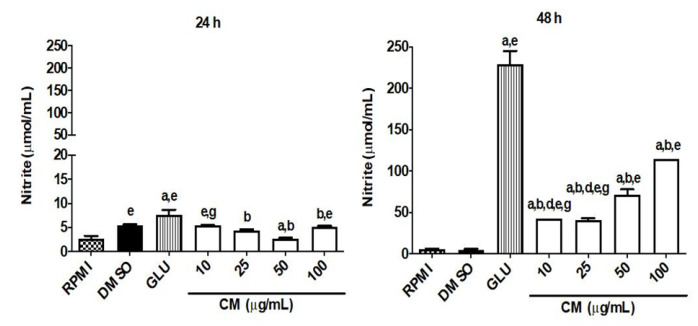
Nitrites levels produced by RAW 264.7 lineage macrophages infected with *L. braziliensis*. Macrophages of the RAW 264.7 lineage (1 × 10^6^ cells/mL) were cultured with *L. braziliensis* (10^7^ promastigotes/mL), in a ratio of 10 parasites to 1 macrophage. After 12 h, they were incubated (37 °C; 5% CO_2_) with CM (10, 25, 50, and 100 µg/mL). After 24 and 48 h, the supernatants were collected for nitric oxide dosages. As a control group, supplemented RPMI medium, DMSO 0.1%, and GLU (4 mg/mL) were used. The absorbance was measured at 560 nm, the nitrite content was determined from a sodium nitrite standard curve, and the results are expressed as μmol nitrite/mL. GLU: glucantime^®^, and CM: coumarin. Each column represents the mean ± S.E.M. *p* < 0.05 to RPMI (a); to 0.1% DMSO (b); to 10 (d); to 25 (e); and to 100 (g) (one-way ANOVA; Tukey test).

**Figure 5 biomedicines-13-00979-f005:**
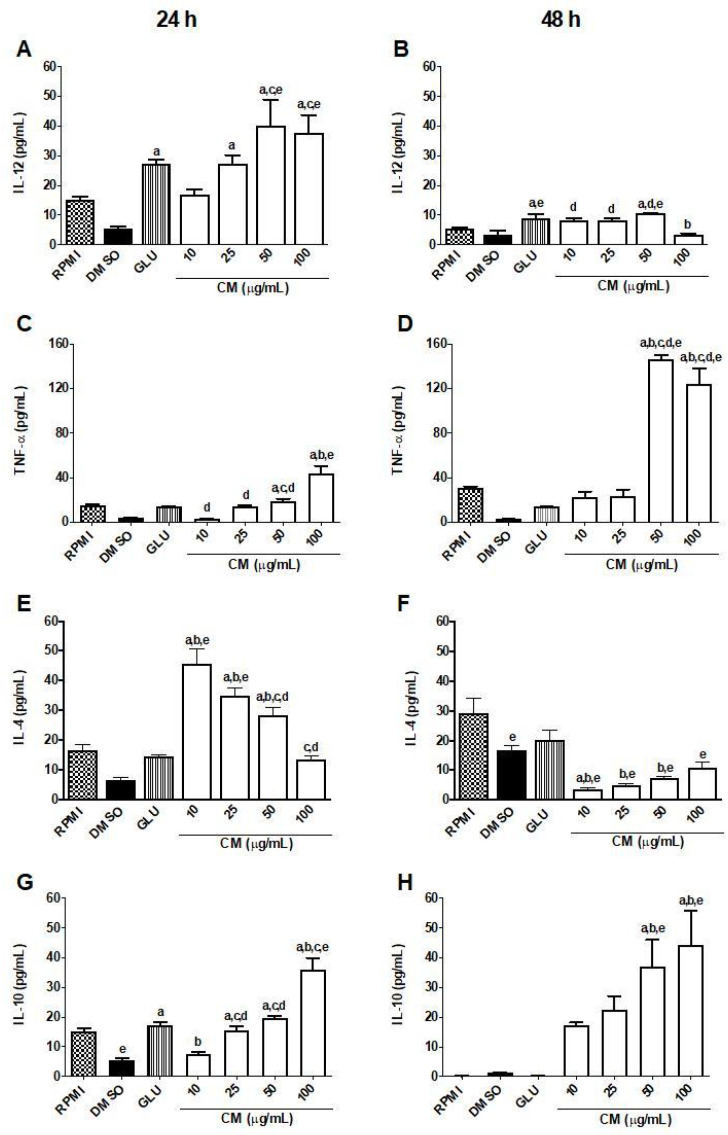
Cytokines levels produced by RAW 264.7 lineage macrophages infected with *L. braziliensis*. Macrophages of the RAW 264.7 lineage (1 × 10^6^ cells/mL) were cultured with *L. braziliensis* (10^7^ promastigotes/mL), in a ratio of 10 parasites to 1 macrophage. After 12 h, they were incubated (37 °C; 5% CO_2_) with CM (10, 25, 50, and 100 µg/mL). After 24 and 48 h, the supernatants were collected for cytokine dosages. As a control group, supplemented RPMI medium, DMSO 0.1%, and GLU (4 mg/mL) were used. The results are expressed as pg/mL. TNF: tumor necrosis factor; IL: interleukin; GLU: glucantime^®^; and CM: coumarin. Each column represents the mean ± S.E.M. *p* < 0.05 to RPMI (a); to 0.1% DMSO (b); to GLU (c); to 10 (d); and to 25 (e) (one-way ANOVA; Tukey test). (**A**) IL-12 levels after 24 h, (**B**) IL-12 levels after 48 h, (**C**) TNF-α levels after 24 h, (**D**) TNF-α levels after 48 h, (**E**) IL-4 levels after 24 h, (**F**) IL-4 levels after 48 h, (**G**) IL-10 levels after 24 h, (**H**) IL-10 levels after 48 h.

## Data Availability

Data are contained within the article.

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
