# Peer review of "Amburana cearensis (Cumaru) and Its Active Principles as Source of Anti-Leishmania Drugs: Immunomodulatory Activity of Coumarin (1,2-Benzopyrone)"

_biomedicines, 2025, doi:10.3390/biomedicines13040979_

Round 1

Reviewer 1 Report

Comments and Suggestions for Authors

To the Authors

  • It is not clear to me if CM is really extracted or is from Sigma.
  • Lines 69-70 I would put effect after antineuroinflammatory
  • Line 70 CM and AMB should be put after their respective names, written two lines before
  • Line 75 put effects after in vitro!!!
  • Line 81 after active you miss the name may be substances
  • Legend of figure 2: As control, supplemented RPMI, pure DMSO and 0.1% and GLU (4 mg/mL) were used. But there is written 10% DMSO
  • Line 255 a full stop is missing
  • Line 264 Put (Fig. 3C) in place of figure 3C, also in lines 366 and 368 Fig. in place of figure
  • Legend of Fig. 3 does not correspond to the figure, in fact there is C, D, E and F, instead of A, B, C and D
  • Line 265 put on the other hand, then cut the word present
  • Line 266 put GLU in place of Glu
  • Fig. 4 While from the figure we see the increase of nitrite for GLU even at 24 h, from the text 7.618 ± 1.058 pg/mL is not an increase of 29% compared to 5.454± 0.2147 pg/mL. Also 229.40 + 15.70 pg/mL is not an increase of 98% compared to 5.454± 0.2147 pg/mL
  • Fig. 4 legend It is better Nitrites levels and not Nitrite’s levels, also I do not see c and f
  • Line 319 The English is not correct, may be you can add Upon before The infection
  • Line 327 Fig 5A and 5B, not 6B
  • Fig. 5 legend : I do not see g. Furthermore, in the Y axis of TNF-a the values do not correspond with what is written in the text: The maximal effect of CM was observed when it was incubated for 48h achieving maximal effect at higher concentration (pg/mL) (50: 18.21 ± 2.887 pg/mL;100: 43.74 ±6.894 pg/mL)
  • Line 335 Fig 5C and 5D, not 6D
  • Line 336 in both the time?? It is not so for 48h!

Discussion

  • Line 366 Cut Authors
  • Line 371 leishmaniose is not English
  • Line 374 correct Leishamania
  • Line 381 replace Worth with worth, moreover I would cut although
  • Line 382 correct rapreseent
  • Line 383 replace forms in the end of the line with from
  • Line 384 correct glycocaly
  • Line 402 “CM from A. cearensis” Do you use CM from Sigma? In this case are you sure they (Sigma) extract that from A. cearensis??
  • In the discussion you write:
    • It is important to emphasize that the result obtained for CM (10 – 100 g/mL) regarding activity against promastigote corroborates with the study carried out by [23], in which the in vitro antileishmanial activity of this metabolite was reported against promastigote forms of Leishamania sp. This ref does not report that, in fact it is the 34 instead.
  • Line 388 it is not ref 36 while it is the 37, probably thus all the refs have wrong numbers

Conclusions

  • It seems to me that conclusions have to be changed writing that CM has a leishmanicidal effect against Leishmania braziliensis (promastigote and amastigote forms), without showing toxicity to macrophages. The effect of DEAC fraction from A. cearensis might be due to CM and DEAC toxicity to macrophages could be due to another extract substance. Anyway, that the leishmanicidal effect of DEAC is due to CM still must be proved.

Comments on the Quality of English Language

It seems to me that English language needs some revision.

Author Response

Manuscript – biomedicines-3497196

Answers to Reviewer’s comments

We thank the reviewers for their valuable commentaries and suggestions. We addressed all queries.

Our responses to each comment are highlighted in blue.

Changes on the manuscript are identified in blue

As requested, the manuscript underwent a thorough revision of the English language.

#Reviewer 1

  1. It is not clear to me if CM is really extracted or is from Sigma.

We appreciate your comments and the opportunity to clarify the raised point. In response to your question about the origin of the coumarin, we have removed the terms that referred to the coumarin extracted from Amburana cearensis. The coumarin used is from Sigma (line 458).

  1. Lines 69-70 I would put effect after antineuroinflammatory

We appreciate your suggestion. As indicated, we have placed the word "effect" after "antineuroinflammatory" (current position: line 72).

  1. Line 70 CM and AMB should be put after their respective names, written two lines before

We appreciate your suggestion. As requested, the acronyms CM and AMB have been added at the first appearance of their respective names, as mentioned two lines earlier (lines 70 and 71).

  1. Line 75 put effects after in vitro!!!

Thank you for your suggestion. As indicated, we have placed the word "effects" after "in vitro" (line 77).

  1. Line 81 after active you miss the name may be substances

In response to your suggestion, the word "materials" has been inserted after "active" (line 83).

  1. Legend of figure 2: As control, supplemented RPMI, pure DMSO and 0.1% and GLU (4mg/mL) were used. But there is written 10% DMSO.

The study used DMSO 10% as cytotoxic standard. So, we removed the term “pure DMSO” (lines 249).

  1. Line 255 a full stop is missing

We added “the full stop” in line 255, as indicated (line 257).

  1. Line 264 Put (Fig. 3C) in place of figure 3C, also in lines 366 and 368 Fig. in place of figure

We have implemented your suggestion, replacing "figure" with "(Fig.)" (lines 261 and 268).

  1. Legend of Fig. 3 does not correspond to the figure, in fact there is C, D, E and F, instead of A, B, C and D

We corrected the legend of Fig. 3, changing it to the correct labeling: A, B, C, and D (lines 284 to 286).

  1. Line 265 put on the other hand, then cut the word present

We appreciate your observation. In response to Reviewer 2's comment, we have reformulated the sentence to make it clearer (lines 264 to 269).

  1. Line 266 put GLU in place of Glu

We have replaced "Glu" with "GLU", as indicated (line 268).

  1. 4 While from the figure we see the increase of nitrite for GLU even at 24 h, from the text 7.618 ±1.058 pg/mL is not an increase of 29% compared to 5.454± 0.2147 pg/mL. Also 229.40 + 15.70 pg/mL is not an increase of 98% compared to 5.454± 0.2147 pg/mL

Thank you for pointing out this inconsistency. We corrected the nitrite increase after 24 h of GLU, and the correct value is 39.7% (line 304). Regarding the increase after 48 h, it was indeed not 98%, so we have adjusted the sentence to: "resulted in a more than 40-fold increase in nitrite level" (line 301).

  1. 4 legend It is better Nitrites levels and not Nitrite’s levels, also I do not see c and f

We have added "Nitrite levels" as suggested. Regarding the statistical test, there was not statistical difference for C (GLU) and F (CM 100 µg/mL), so we removed this information from the legend. Additionally, we noticed that the statistical comparison for D (CM 10 µg/mL) was missing, so we have included it in the legend (lines 310 and 317).

  1. Line 319 The English is not correct, may be you can add Upon before The infection

We added the word "Upon" before "The infection" (line 322).

  1. Line 327 Fig 5A and 5B, not 6B

We corrected the reference to Fig. 5A and 5B, instead of 6B, as indicated (line 329).

  1. 5 legend: I do not see g

It was mistake. There was not statistical difference when compared CM 100 µg/mL group versus the other groups. So, we removed the letter "g" from the legend of Fig. 5, (line 361).

  1. Furthermore, in the Y axis of TNF-α the values do not correspond with what is written in the text: The maximal effect of CM was observed when it was incubated for 48h achieving maximal effect at higher concentration (pg/mL) (50: 18.21 ± 2.887 pg/mL;100: 43.74 ±6.894 pg/mL)

We appreciate your observation. The error has been corrected, and the values for the maximal effect at the highest concentrations have been adjusted to (50: 146 ± 4.01 pg/mL; 100: 124 ± 14.24 pg/mL) (lines 333 and 334).

  1. Line 335 Fig 5C and 5D, not 6D

We noticed the error and have corrected it to properly refer to Figures 5C and 5D, instead of 6D (line 337).

  1. Line 336 in both the time?? It is not so for 48h! Discussion

Thank you for the comment. Indeed, no statistical difference was observed in the IL-4 increase at 48 hours. A significant increase was only observed at 24 hours, and therefore, we have removed the description regarding 48 hours in the statistical description section (lines 338).

  1. Line 366 Cut Authors

We removed the word 'authors' from the beginning of the text, as suggested (line 365).

  1. Line 371 leishmaniose is not English Line 374 correct Leishamania

We have made the correction, changing leishmaniose to leishmaniasis and correcting Leishamania to Leishmania (lines 370 and 373).

  1. Line 381 replace Worth with worth, moreover I would cut although Line 382 correct rapresent

We made the recommended changes (line 380).

  1. Line 383 replace forms in the end of the line with from Line 384 correct glycocaly

We make the corrections (lines 382 and 383).

  1. Line 402 “CM from cearensis” Do you use CM from Sigma? In this case are you sure they (Sigma) extract that from A. cearensis??

We rewrote the text as follows: In the present work, CM demonstrated an anti-amastigote effect. The coumarin was obtained from Sigma and not isolated from Amburana cearensis (line 402).

  1. In the discussion you write:

It is important to emphasize that the result obtained for CM (10 ‒ 100 g/mL) regarding activity against promastigote corroborates with the study carried out by [23], in which the in vitro  antileishmanial activity of this metabolite was reported against promastigote forms of Leishamania sp.

This ref does not report that, in fact it is the 34 instead

The reference 34 (title: 'Antileishmanial activity and immune modulatory effects of benzoxonium chloride and its entrapped forms in niosome on Leishmania tropica) is not related to coumarin.  After revision the reference 34 became 33. The correct reference is titled “Natural and synthetic coumarins as antileishmanial agents: a review” (reference number 27) (line 372).

  1. Line 388 it is not ref 36 while it is the 37

Probably thus all the refs have wrong numbers

We conducted a review of the references, and the correct reference was added. (line 387).

  1. Conclusions

It seems to me that conclusions have to be changed writing that

CM has a leishmanicidal effect against Leishmania braziliensis (promastigote and amastigote forms), without showing toxicity to macrophages. The effect of DEAC fraction from A. cearensis might be due to CM and DEAC toxicity to macrophages could be due to another extract substance. Anyway, that the leishmanicidal effect of DEAC is due to CM still must be proved.

We rewritten the conclusion (lines 460-464).

Reviewer 2 Report

Comments and Suggestions for Authors
  1. Abstract: How musch inhibition due to CM was observed to be indicated in the results section, before putting as promising in the conclusion.
  2. Line 19 - reword for clarity
  3. 3.Line 21:selection of active ...? from, please add appropriate word
  4. Authors may use another acronym for amburoside A in place of AMB, as it is a common acronym used for Amphotericin B.
  5. Line 74,: rephrase for larity
  6. Fig legen ANF B for amphotericin B to be replaced with AmB
  7. Line 262-265: reword
  8. Figure legend line 284-286. e and picture not available
  9. 3.3.1. oxide production in .... Please reword
  10. 298: reword
  11. Line 365
  12. line 12: either write author after by or use earlier study
  13. line 458 active pharmaceutical ingredients...

Comments on the Quality of English Language

The language needs to be checked throughout the manuscript for clarity. The attached manuscript highlights a few instances.

Author Response

Manuscript – biomedicines-3497196

Answers to Reviewer’s comments

We thank the reviewers for their valuable commentaries and suggestions. We addressed all queries.

Our responses to each comment are highlighted in blue.

Changes on the manuscript are identified in blue

As requested, the manuscript underwent a thorough revision of the English language.

#Reviewer 2

  1. Abstract: How much inhibition due to CM was observed to be indicated in the results section, before putting as promising in the

The inhibition percentages of the promastigote and amastigote forms of L. braziliensis by coumarin were added to the results section of the abstract, as well as those of the reference drug, Glucantime.

  1. Line 19 - reword for clarity

As suggested, the text has been rewritten, and the changes are highlighted.

  1. Line 21: selection of active ...? from, please add appropriate word

As recommended, the word has been changed in the text (Line 22) as    follows:

The present study aimed to carry out the bioprospecting and selection of products of Amburana cearensis, including extract and active principles with leishmanicidal...

  1. Authors may use another acronym for amburoside A in place of AMB, as it is a common acronym used for Amphotericin B.

The acronym of the amburoside A was changed to AMR.

  1. Line 74,: rephrase for clarity

The text (line 76) was rephrase as follows:

A. cearensis and/or its chemical constituents, including antioxidant [17,18], muscle relaxant [19], antimicrobial [20,21] and antimalarial activities [22].

  1. Fig legend ANF B for amphotericin B to be replaced with AmB

As recommended, we have changed the acronym ANF B to ANF (lines 227-232).

  1. Line 262-265: reword

The paragraph has been rewritten (lines 264-269).

  1. Figure legend line 284-286 and picture not available

The groups of the figure 3 were incorrectly identified; in fact, they correspond to figures A, B, C, and D, not figures E, F, G, and H, as they were previously labeled (lines 285-286).

In addition, we improve the quality of images (figure 3A, B, C, and D).

  1. Item 3.3.1. oxide production in….. Please reword

As suggested, the text was corrected, as follows:

3.3.1 CM induces an increase in nitric oxide production and modulates the cytokine profile in macrophages infected by L. braziliensis (line 291 and 292).

  1. Line 298: reword

The text was reworded, as follows:

“We corrected the nitrite increase after 24 h of GLU, and the correct value is 39.7% (line 304). Regarding the increase after 48 h, it was indeed not 98%, so we have adjusted the sentence to: "resulted in a more than 40-fold increase in nitrite level" (line 300 and 301).

  1. Line 365

We don´t understand the question. If there is something specific that needs adjustments, please let us know.

  1. line 12: either write author after by or use earlier study

Dr. Alda Maria da Cruz provided the strains of L. braziliensis (line 124).

  1. line 458 active pharmaceutical ..

The text was included as recommended (line 458).

Additional information:

We conducted a careful review of the references in the manuscript and, we noticed that reference 12 (item Reference) was not cited in text, so it was removed.
